# *DPA4* Suppresses Adventitious Root Formation via Transcriptional Regulation of *CUC2* and *ULT1*, Decreasing Auxin Biosynthesis in Arabidopsis Leaf Explants

**DOI:** 10.3390/ijms262311336

**Published:** 2025-11-24

**Authors:** Yucai Zheng, Qian Xing, Xuemei Liu, Ralf Müller-Xing

**Affiliations:** 1College of Life Science, Northeast Forestry University, Harbin 150040, China; zhengyucai@nefu.edu.cn (Y.Z.); liuxuemei@nefu.edu.cn (X.L.); 2Jiangxi Provincial Key Laboratory of Plant Germplasm Resource Innovation and Genetic Improvement, Lushan Botanical Garden, Chinese Academy of Sciences, Jiujiang 332900, China; qxing@lsbg.cn; 3Plant Epigenetics and Development, Lushan Botanical Garden, Chinese Academy of Sciences, Nanchang 330114, China; 4College of Life Science, Nanchang University, Nanchang 330047, China

**Keywords:** de novo root organogenesis (DNRR), adventitious roots (ARs), auxin biosynthesis, *YUCCA* (*YUC*) genes, *DPA4*, *CUC2*, *ULT1*

## Abstract

Plants have the capacity to form adventitious roots (ARs) from detached aerial organs, a process known as de novo root regeneration (DNRR). In Arabidopsis, wounding signals rapidly induce in leaf explants the expression of genes encoding enzymes of auxin biosynthesis, resulting in elevated auxin levels and facilitating AR formation. Here, we report that DEVELOPMENT-RELATED POLYCOMB TARGET IN THE APEX 4 (DPA4/NGAL3), a well-known regulator in seed size and leaf margin development, and a repressor of *CUP-SHAPED COTYLEDON 2* (*CUC2*), inhibits AR formation in detached leaves. Leaf explants of *dpa4-2* and *cuc2-1D* mutants displayed both elevated *CUC2* mRNA levels and increased rooting rates. We observed reduced expression of *ULTRAPETALA1* (*ULT1*), a negative regulator of DNRR, while the auxin biosynthesis genes *ASA1*, *YUC4*, and *YUC9* were upregulated in both mutants. Through pharmacological inhibition of YUCCA-mediated auxin biogenesis, we obtained evidence that the enhanced AR formation in both mutants is at least partially a result of increased auxin production. Genetic analysis of *dpa4-2 cuc2-1D* double mutants indicates that similar mechanisms promote DNRR in both mutants. In summary, our study suggests that *DPA4* suppresses AR formation likely by repression of *CUC2* and activation of *ULT1*, which, in turn, suppresses endogenous auxin biogenesis and DNRR.

## 1. Introduction

Plant cells exhibit remarkable developmental plasticity, enabling regeneration of new organs and even an entire plant body after suffering wounds [1,2]. The potential capacity of plants for de novo organogenesis finds wide-ranging applications in agriculture, biotechnology, and biological studies, including tissue culture and vegetative propagation via cuttings and explants [3,4]. Among various forms of plant regeneration, de novo root regeneration (DNRR) from wounded and/or detached plant tissues and organs is commonly utilized in biotechnological breeding and cultivation research [5,6]. In contrast to de novo shoot regeneration that requires a proper ratio of exogenous auxin to cytokinin [7,8], DNRR from Arabidopsis leaf explants can take place on B5 medium without exogenous phytohormones [9,10].

Upon injury, Arabidopsis leaf explants sense a multitude of signals, including wound-derived signals and other stress-related and environmental stimuli, factors triggering endogenous developmental programs [11,12,13,14]. These signals direct the production and transport of auxin towards the future regeneration-competent cells near the wound site, which facilitates adventitious root (AR) formation [15,16]. Once the leaf is detached, the wounding signal and phytohormone jasmonate (JA) promptly trigger auxin biosynthesis [12,17]. Therefore, JA signaling activates the expression of *ETHYLENE RESPONSE FACTOR 109* (*ERF109*) and *ABSCISIC ACID REPRESSOR 1* (*ABR1/ERF111*) [11,12]. The transcriptional activators ERF109 and ABR1 serve as a molecular connection between wounding, JA signaling, and auxin biosynthesis by direct activation of *ANTHRANILATE SYNTHASE ALPHA SUBUNIT 1* (*ASA1*) [12,18]. ASA1 catalyzes the rate-limiting step in tryptophan (Trp) biosynthesis, which provides the substrate for auxin biosynthesis. Via the conversion of Trp to indole-3-pyruvate (IPA) by the TAA amino transferases, YUCCA (YUC) flavin-containing monooxygenases convert IPA to auxin indole-3-acetic acid (IAA), one of the major natural auxins [19,20,21]. The newly synthesized auxin is polar-transported to the future rooting site adjacent to the wounding site [22]. The emerging auxin maximum induces the expression of *WUSCHEL-RELATED HOMEOBOX 11* (*WOX11*)/*WOX12*, which facilitates the cell fate transition from regeneration-competent vasculature-associated pluripotent cells (VPCs) to root founder cells [23,24,25]. Both *WOX11/12* and auxin are required to promote the expression of *LATERAL ORGAN BOUNDARIES DOMAIN 16* (*LBD16*) and *WOX5*, which facilitates the transformation of the root founder cells into root primordia [24,26,27]. Disrupting either auxin biosynthesis or transport prevents the establishment of the auxin maxima in the VPCs and, in turn, AR formation [28].

In Arabidopsis, *CUP-SHAPED COTYLEDON2* (*CUC2*) encodes an NAC domain transcription factor that is a key regulator that establishes organ boundaries and participates in multiple developmental pathways [29]. In leaves, *CUC2* is specifically targeted by *MIR164A*, which triggers the cleavage of the *CUC2* mRNA, while *cuc2-1D*, which carries a point mutation in the *MIR164A*-targeting site, and *mir164a-4* mutants display increased *CUC2* expression [30,31]. During leaf margin patterning, *CUC2* establishes auxin activity maxima by directing PIN1 localization at the leaf margin, thereby shaping serrated leaf morphology [32,33]. Furthermore, *CUC2* restricts leaf expansion primarily through modulating cell proliferation rather than influencing cell expansion [34].

Notably, *CUC2* expression is indispensable for de novo shoot organogenesis from callus tissue [35,36,37] and serves as a molecular indicator of regenerative potential in root-derived explants [38]. Collectively, these findings establish *CUC2* as a key regulator of auxin distribution patterns. Since auxin is essential for AR formation [28], it is a reasonable question whether *CUC2* has a function in DNRR from leaf explants.

The B3 transcription factor DEVELOPMENT-RELATED POLYCOMB TARGET IN THE APEX4 (DPA4/NGAL3), belonging to the RAV (Related ABI3/VP1) family, has been implicated in multiple developmental processes, including leaf margin development [39,40], de novo stem cell formation in axillary meristems [32], and seed size regulation [41,42]. *DPA4* is expressed in the shoot apex during primordia formation, while also being expressed in the leaf sinuses coinciding with the *CUC2* expression domain [30,39]. Genetic studies demonstrate that *DPA4* suppresses *CUC2* expression to regulate leaf morphogenesis [39,40]. Notably, *DPA4* expression is induced early by wounding in leaf explants during DNRR [11]. However, the regulatory role of *DPA4* in DNRR remains to be elucidated.

A previous study showed that *DPA4* belongs to a group of transcription factor genes that are induced early by wounding during DNRR [11]. However, it remained unclear whether *DPA4* has positive or negative effects on wound-induced DNRR. In this study, we reveal that *DPA4*, encoding a well-known repressor of *CUC2* expression, inhibits DNRR from *Arabidopsis* leaf explants. Besides the loss-of-function mutant *dpa4-2*, we employed two other *CUC2*-overexpressing lines, *cuc2-1D* and *mir164a-4*, to test whether increased *CUC2* function promotes DNRR in general. All three mutants displayed enhanced AR formation, suggesting that *CUC2* promotes DNRR downstream of *DPA4*. Furthermore, we found similar expression changes in *dpa4-2* and *cuc2-1D* mutant leaf explants, which include increased mRNA levels of *CUC2*, *ERF109*, *ASA1*, and *YUC4/9*, while *ULT1* was lower-expressed compared to the wild-type. The enhanced root regeneration was associated with increased auxin concentrations in both *cuc2-1D* and *dpa4-2* mutant leaf explants, as indicated by intensified *DR5::GUS* reporter activity, especially in and around the rooting side. Genetic analysis of *dpa4-2 cuc2-1D* double mutants and pharmacological inhibition of YUC-mediated auxin biogenesis supported the hypothesis that increased auxin production and enhanced DNRR are causally linked. These results indicate a mechanistic link between transcriptional regulation of *CUC2* by DPA4 and auxin-dependent developmental reprogramming during DNRR. Our work provides new insights into the molecular control and gene regulatory network of AR formation and offers potential biotechnological applications for improving vegetative propagation of crop plants.

## 2. Results

### 2.1. DPA4 Suppresses De Novo Root Regeneration (DNRR) from Leaf Explants

To investigate the potential role of *DPA4* in AR formation, we conducted DNRR assays following established protocols [9], in which detached leaf explants were cultured on phytohormone-free B5 medium under dark conditions. We quantified the rooting rate and root regeneration capacity in leaf explants of the loss-of-function mutant *dpa4-2* and wild-type (Col-0) plants (Figure 1A–D). The time-course analysis revealed that *dpa4-2* leaf explants exhibited significantly accelerated root formation rates compared to the wild-type control (Figure 1A–D). Furthermore, the rooting capacity was significantly increased in *dpa4-2* mutant leaf explants in comparison with the wild-type (Figure 1D). These findings indicate that *DPA4* plays a role in suppressing DNRR in Arabidopsis.

### 2.2. DPA4 Regulates Genes Known to Be Involved in AR Formation and Auxin Biosynthesis

Since *DPA4* encodes a transcription factor, we investigated the expression changes of putative downstream targets of *DPA4* after wounding to better understand its function in suppressing DNRR. For the expression analysis of the candidate genes, we harvested leaf explants at 4 h after culturing (HAC) on B5 medium. Since *DPA4* is a known repressor of the boundary gene *CUC2* in leaf margin development [39], we examined *CUC2* expression levels in leaf explants. The reverse transcription quantitative PCR (RT-qPCR) analysis revealed significantly increased *CUC2* expression in the *dpa4-2* mutant compared to the wild-type (Figure 2), confirming that *DPA4* functions upstream of *CUC2* in leaf explants (4 HAC). In a previous study, we found that *ULT1* is a negative regulator of DNRR, likely by repressing *ERF109* and, in turn, the auxin biosynthesis gene *ASA1* [43]. Since loss of *ULT1* phenocopies the acceleration of AR formation in *dpa4-2* mutants, we analyzed the expression of *ULT1*, *ERF109*, and *ASA1* (Figure 2). Our quantitative analysis revealed significantly reduced levels of *ULT1* mRNA, while the expression of *ERF109* and *ASA1* was increased in *dpa4-2* leaf explants compared to the wild-type. Like in *ult1* mutants, the expression of *ABR1*, another wounding-induced activator of *ASA1*, was not significantly changed in *dpa4-2* (Figure 2). These results suggest that *DPA4* constrains AR formation at least partially by activation of *ULT1* that, in turn, represses *ERF109* and thereby limits auxin biosynthesis through the direct repression of *ASA1* by ERF109. The expression levels of *YUC* genes have been reported as the rate-limiting step in auxin biosynthesis and DNRR [28,44,45]. Therefore, we examined the expression levels of *YUC* genes in *dpa4-2* leaf explants, 4 HAC (Figure 2). Notably, *YUC4*, *YUC6*, and *YUC9* showed significant upregulation in *dpa4-2* compared to the wild-type, while *YUC1* expression was not significantly changed. Furthermore, the auxin-inducible DNRR-factor *WOX11* [23,24] exhibited significant upregulation in *dpa4-2* (Appendix A). Collectively, these results indicate that the enhanced rooting rate and rooting capacity observed in *dpa4-2* mutant leaf explants are likely caused by an increased auxin biosynthesis.

### 2.3. cuc2-1D Phenocopies the Increased AR Formation Phenotype of dpa4-2 Mutant Leaf Explants, While Both Mutants Display Similar Expression Changes in DNRR-Related Genes

Since *CUC2* is a known target of *DPA4* in leaf margin development [39] and plays an essential role in de novo shoot regeneration [35,36,37], we hypothesized that the increased *CUC2* expression might promote AR formation in *dpa4-2* mutants. To test whether increased *CUC2* expression levels can promote DNRR, we employed *cuc2-1D* mutants that carry a mutation in the *CUC2* miRNA target site, resulting in *CUC2* overexpression [31]. We performed DNRR assays to check the rooting rate and capacity in *cuc2-1D* mutant leaf explants (Figure 3A,E,G). Furthermore, we tested the rooting rate and capacity in *mir164a-4* mutant leaf explants (Appendix A). Consistent with our hypothesis, *cuc2-1D* and *mir164a-4* mutant leaf explants exhibited similar increased rooting rates and rooting capacity as *dpa4-2* mutants, which were significantly higher than those of the wild-type, suggesting that increased *CUC2* expression is sufficient in promoting AR formation.

Since *cuc2-1D* phenocopies the increased AR formation phenotype of *dpa4-2* mutants, we analyzed the expression of genes that was misregulated in *dpa4-2* in *cuc2-1D* mutant leaf explants during DNRR (Figure 2). Like in *dpa4-2* mutants, *ULT1* was downregulated, while *CUC2*, *ERF109*, *ASA1*, *YUC4*, *YUC6*, and *YUC9* were upregulated in *cuc2-1D* mutants in comparison to the wild-type, indicating that increased auxin levels could cause the increased AR formation phenotype in both mutants. Notably, also in *miR164a-4* mutant leaf explants, *CUC2* was upregulated and *ULT1* was downregulated (Appendix A). Although *ASA1* was upregulated in all three *CUC2*-overexpressing mutants, partially different *YUC* genes were upregulated in *dpa4-2*, *cuc2-1D*, and *mir164a-4* (Figure 2 and Appendix A), indicating that increased *CUC2* is not the only factor that influences *YUC* expression in these three lines. Nevertheless, the auxin-inducible DNRR-factor *WOX11* [23,24] exhibited significant upregulation in *dpa4-2*, *cuc2-1D*, and *mir164a-4,* indicating increased auxin biosynthesis (Appendix A).

To support the hypothesis that increased auxin levels cause the enhanced DNRR in *cuc2-1D* and *dpa4-2* mutants, we conducted DNRR assays that either increased the auxin levels by exogenous application of IAA or decreased endogenous IAA levels in the leaf explant by applying the auxin biosynthesis inhibitor yucasin (Figure 3). Our treatment with 0.1 μM IAA accelerated AR formation in all three genotypes without equalizing the rooting rates [43], and *cuc2-1D* and *dpa4-2* leaf explants displayed earlier rooting initiation compared to the wild-type, although the differences in the time-course were reduced relative to mock (Figure 3E,F). Furthermore, the rooting capacity was equalized in wild-type, *cuc2-1D*, and *dpa4-2* leaf explants (Figure 3G), corroborating the idea that accelerated and enhanced AR formation is mainly caused by increased auxin levels in both mutants.

Next, we tested the influence of increased and decreased gibberellin (GA) levels on DNRR in *dpa4-2* and *cuc2-1D* mutant leaf explants. The phytohormone GA is well-known for its negative effects on AR formation, likely through its impact on auxin transport [27,46]. Although the changes in rooting capacity and the overall effects of the GA synthesis inhibitor PBZ were less distinct, exogenous GA significantly decreased the rooting rates in Col-0, *cuc2-1D*, and *dpa4-2* leaf explants, while the differences in the rooting rate between Col-0 and both mutants with increased *CUC2* levels remained (Appendix A), indicating that GA signaling is unaffected during DNRR in *cuc2-1D* and *dpa4-2* mutants.

Since the expression of several *YUC* genes is upregulated in *dpa4-2* and *cuc2-1D* mutant leaf explants, we investigated whether *YUC*-mediated auxin biosynthesis is required for the enhanced AR formation in both mutants. To test this, we used yucasin, which is a specific inhibitor of the YUC enzymes, reducing endogenous auxin levels during DNRR on B5 medium [28,47]. At a low concentration of 30 μM yucasin, DNRR in wild-type, *cuc2-1D*, and *dpa4-2* leaf explants was partially reduced, resulting in decelerated rooting rates over the time-course and lower rooting capacity (Figure 3C,H,J). As expected, combined treatment with 30 μM yucasin and 0.1 μM IAA partially rescued the rooting rate and rooting capacity in all three lines (Figure 3D,I,J). Notably, the combined treatment with yucasin and IAA equalized the rooting rate and rooting capacity in wild-type and *dpa4-2* leaf explants, strongly suggesting that the enhanced rooting activity in *dpa4-2* leaf explants is primarily caused by increased YUC-dependent auxin biosynthesis.

In contrast, the rooting rate and rooting capacity of *cuc2-1D* mutant leaf explants remained significantly higher than those of the wild-type during low (30 μM)-yucasin treatments with and without IAA (Figure 3C,D,I,J). This could be caused by higher auxin levels in *cuc2-1D* than in *dpa4-2* leaf explants or an auxin-independent factor that promotes increased DNRR in *cuc2-1D*, but not in *dpa4-2*. To rule out the latter one, we repeated the DNRR experiment with a higher concentration of 100 μM yucasin with and without 0.1 μM IAA (Figure 3K–M). The single treatment with 100 μM yucasin annulled the rooting activity in all three lines, confirming that YUC-dependent auxin biosynthesis is, in general, essential for AR formation (Figure 3K,M). The addition of IAA partially rescued the disrupted rooting activity caused by 100 µM yucasin, and the low rooting rates and rooting capacities were not significantly different among all three genotypes (Figure 3L,M). In summary, these findings indicate that the enhanced DNRR ability in *cuc2-1D* and *dpa4-2* mutants results from elevated auxin production. This is in line with the hypothesis that *DPA4* regulates DNRR through a *CUC2*-dependent pathway that controls *YUC*-mediated auxin biosynthesis.

### 2.4. DPA4 and CUC2 Promote Endogenous Auxin Levels

To confirm that the enhanced DNRR activity in *dpa4-2* and *cuc2-1D* results from high auxin production in leaf explants during regeneration, we performed a GUS reporter gene assay using the *DR5::GUS* reporter to show the spatiotemporal pattern of auxin in the leaf explants. We analyzed *DR5::GUS* staining in wild-type, *dpa4-2*, and *cuc2-1D* leaf explants at two time points (0 DAC and 1 DAC) (Figure 4). The *DR5::GUS* reporter activity reflects free auxin levels [48]. As expected, the *DR5::GUS* expression was low and absent in the potential rooting side in all three genotypes at 0 DAC (Figure 4A–C) [24,28]. After leaf explants were placed on B5 medium (mock) for one day (1 DAC), *DR5::GUS* staining intensively increased above the wounding sites and was significantly stronger in *cuc2-1D* and *dpa4-2* leaf explants compared to the wild-type, while the staining was stronger in *cuc2-1D* than *dpa4-2* (Figure 4D–F). At 1 DAC, the differences in *DR5::GUS* expression between *dpa4-2* and *cuc2-1D* remained evident during all treatments with 0.1 μM IAA (Figure 4G–I), 100 μM yucasin (Figure 4J–L), and 100 μM yucasin and 0.1 μM IAA (Figure 4M,N), though the staining was always much weaker in the wild-type. As expected, the IAA treatments increased, while yucasin decreased the *DR5::GUS* staining intensity, and the yucasin + IAA double treatment rescued the *DR5::GUS* staining near mock levels. Taken together, the differences in the intensity of the *DR5::GUS* staining (Figure 4) correlate largely with the previously detected differences in rooting rate and rooting capacity caused by the different treatments and genotypes (Figure 3). This strongly supports the hypothesis that increased auxin levels at the rooting side, likely caused by increased auxin biosynthesis, result in the enhanced DNRR activity in *cuc2-1D* and *dpa4-2* leaf explants.

### 2.5. DPA4 and CUC2 Regulate AR Formation Likely Through a Common Genetic Pathway That Controls Auxin Biosynthesis

Given that *CUC2* is overexpressed in both *cuc2-1D* and *dpa4-2* (Figure 2), we investigated whether they function in the same genetic pathway by generating *cuc2-1D dpa4-2* double mutants. We subsequently examined the rooting rate and capacity in leaf explants from wild-type, *cuc2-1D*, *dpa4-2*, and *cuc2-1D dpa4-2* double mutants (Figure 5A–C). We found that the *dpa4-2* and *cuc2-1D* single mutants and the *cuc2-1D dpa4-2* double mutants exhibited a similar increase in rooting rate and capacity compared to the wild-type; no additive or synergistic effect could be detected. Furthermore, we examined the expression of genes related to DNRR in the *cuc2-1D dpa4-2* double mutant leaf explants using RT-qPCR (Figure 5D). We found that *ULT1* was reduced to the same level in *cuc2-1D*, *dpa4-2,* and *cuc2-1D dpa4-2* leaf explants. In contrast, the expression of *ERF109*, *YUC4*, and *YUC9* was significantly upregulated in *cuc2-1D*, *dpa4-2*, and *cuc2-1D dpa4-2* compared to the wild-type. Moreover, the expression levels of *ULT1*, *ERF109*, *YUC4*, and *YUC9* in the *cuc2-1D dpa4-2* double mutants remained similar to those in *cuc2-1D* and *dpa4-2* single mutants, consistent with the observation that the *cuc2-1D dpa4-2* double mutants did not show enhanced DNRR compared to the *cuc2-1D* and *dpa4-2* single mutants. Interestingly, *ASA1* expression, which was increased in the *cuc2-1D* and *dpa4-2* single mutants in a previous experiment (Figure 2), was only significantly increased in the *cuc2-1D dpa4-2* leaf explants (Figure 5D). This might reflect that *ASA1* expression levels, which are only slightly increased in *ult1-3*, *cuc2-1D*, *dpa4-2*, and *cuc2-1D dpa4-2* double mutants (Figure 2 and Figure 5D) [43], are not solitarily dependent on the expression levels of *ERF109* but on variable environmental factors. In conclusion, the absence of additive or synergistic effects in *cuc2-1D dpa4-2* double mutants on the rooting phenotype and expression patterns strongly supports that *DPA4* and *CUC2* function in the same genetic pathway that controls the expression of *ULT1* and auxin biosynthesis genes regulating DNRR in leaf explants.

## 3. Discussion

DNRR is a developmental process that enables plants to regenerate new roots post injury, after detachment, and under other stress conditions, representing a crucial survival strategy of higher plants [9,15]. Mechanical wounding primarily drives this process of AR formation by creating high auxin concentrations near the wounding site [28]. Despite its importance, the molecular pathways coordinating DNRR, which include hormone responses and the control of intrinsic developmental programs by transcription and epigenetic factors [15,16], are poorly characterized.

A previous study showed that *DPA4* belongs to a group of transcription factor genes that are induced early by wounding during DNRR, but it remained unclear whether *DPA4*, which belongs to the B3 transcription factor gene subfamily, has positive or negative effects on wound-induced DNRR [11]. In this study, we investigated the role of *DPA4* in AR formation in a series of DNRR assays and found that *DPA4* is a suppressor of root regeneration from leaf explants via reduced auxin content. Therefore, *DPA4* likely controls auxin biosynthesis through the indirect activation of *ULT1*, a previously known suppressor of DNRR [43], and the repression of *CUC2*, a newly identified promoter of DNRR (Figure 6).

During screening of candidate genes in DNRR assays, we observed significantly enhanced AR formation in *dpa4-2* mutant leaf explants compared to wild-type (Figure 1). This finding suggests that *DPA4* functions as a key regulator of root regeneration, supporting the emerging view that plant B3 transcription factors play important roles in regulating DNRR [11,17,49]. Our RT-qPCR analysis confirmed increased *CUC2* expression in *dpa4-2* leaf explants, consistent with previous reports showing that *CUC2* is a downstream target of *DPA4* in various developmental processes [32,39,40]. Furthermore, we detected enhanced expression of JA-induced *ERF109*, a member of the AP2/ERF transcription factor subfamily that mediates crosstalk between JA signaling and auxin biosynthesis [12,18]. These observations suggest that *DPA4* may regulate DNRR through modulation of auxin biosynthesis.

Furthermore, we found that several *YUC* genes were significantly upregulated in *dpa4-2*, *cuc2-1D*, and *miR164a-4* mutants that all displayed increased *CUC2* expression (Figure 2 and Appendix A). This result strongly suggests that increased *CUC2* levels are sufficient in promoting DNRR by activating the expression of various *YUC* genes, corroborating previous findings that *YUC1* and *YUC4* expression depend on *CUC2* during embryogenesis [50]. We found that *ULT1*, which encodes a known negative regulator of DNRR [43], was equally downregulated in all three *CUC2* overexpression mutants, *dpa4-2*, *cuc2-1D*, and *miR164a-4* (Figure 2 and Appendix A), suggesting that CUC2 is a direct or indirect repressor of *ULT1* expression (Figure 6). Since ULT1 is a general repressor of *ERF109* [51] that encodes an activator of *ASA1* during DNRR [12,43], the low *ULT1* expression levels may contribute to the high auxin levels via *ASA1*-dependent auxin biosynthesis (Figure 6). However, since the increase in *ASA1* expression is weak and variable, *ULT1* may suppress DNRR also through *ASA1*-independent pathways. Notably, the enhanced root regeneration phenotype of *cuc2-1D dpa4-2* double mutants was very similar to that of the single mutants, and no additive or synergistic effect could be detected (Figure 5), indicating that *DPA4* and *CUC2* function mainly in the same DNRR pathway (Figure 6).

Auxin levels play a pivotal role in DNRR. Previous studies have demonstrated that both exogenous IAA application and inhibition of auxin production by yucasin significantly affect the root regeneration process [28]. Our findings confirmed that increased auxin biosynthesis is essential for the enhanced DNRR in *dpa4-2* and *cuc2-1D* mutants. This conclusion is further supported by our analysis of the *DR5::GUS* expression patterns in leaf explants in DNRR assays with and without IAA and/or yucasin treatment (Figure 4). The intensity of the *DR5::GUS* staining correlates largely with the differences in rooting rate and rooting capacity caused by the different treatments and genotypes (Figure 3). Furthermore, the auxin-inducible DNRR factor *WOX11* exhibited significant upregulation in *dpa4-2*, *cuc2-1D*, and *miR164a-4* (Appendix A), indicating a link between increased *CUC2* expression and auxin biosynthesis, but more importantly, an immediate increase in a key regulator of cell fate reprogramming during DNRR, *WOX11* [23,24]. Nevertheless, we cannot exclude the possibility that *CUC2* and *DPA4* regulate root regeneration also through other pathways [12,52,53].

In summary, our findings suggest that *DPA4* plays a crucial role in DNRR in Arabidopsis leaf explants by serving as a key player in inhibiting *ASA1*- and *YUC*-mediated auxin biosynthesis. Furthermore, *DPA4* and *CUC2* regulate DNRR through a common auxin-dependent pathway. In our conceptual model (Figure 6), *DPA4* suppresses DNRR by downregulating *CUC2*, which, in turn, upregulates *ULT1* that represses *ERF109* and, ultimately, *ASA1*. Likely independently of *ULT1*, *CUC2* represses *YUC*-mediated auxin biosynthesis. Overproliferation of ARs would decrease the survival rate of detached aerial organs [43]. Therefore, the DPA4-CUC2-ULT1 module likely evolved to limit DNRR, thereby increasing the persistence of detached Arabidopsis leaves.

## 4. Materials and Methods

### 4.1. Plant Materials and Culturing Conditions

All plant materials were in the Col-0 background. Seeds of the T-DNA insertion mutant *dpa4-2* [39] (N650707) and the miRNA-resistant *cuc2-1D* allele [31] (N16485) were obtained from the Nottingham Arabidopsis Stock Center. *DR5::GUS* were kindly provided by Rüdiger Simon. The seeds were surfaced-sterilized by three sequential treatments with 70% ethanol (3 min each), and then the seeds were sown on half-strength Murashige and Skoog (1/2 MS) plates containing 1% sucrose and 1% agar with pH adjusted to 5.8. After sowing, the plates were first kept at 4 °C for 3 days in darkness, and then transferred to growth chambers maintained at 22 °C under long-day photoperiod conditions (16 h light/8 h dark cycle).

### 4.2. De Novo Root Regeneration (DNRR) Assays with and Without Hormone Treatment

The first pair of primary leaves was cut from 12-day-old Arabidopsis seedlings for the DNRR assay. Leaf explants were cultured following previously described methods [9] on B5 medium (Gamborg B5 medium containing 3% sucrose, 1% agar, and 0.5 g/L MES, with pH adjusted to 5.7) under dark conditions. The rooting rate was calculated as the percentage of leaf explants developing roots at a given time point [9], while regeneration capacity was assessed based on the percentage of leaf explants showing varying numbers of regenerated roots [11]. For IAA and yucasin treatments, we used 0.1 µM IAA (Sigma-Aldrich, Inc., St. Louis, MO, USA) and 30 µM or 100 µM yucasin (MACKLIN, Beijing, China) in B5 medium (applied continuously after detachment). Each experiment included more than five plates per treatment, with at least eight explants per genotype per plate. A one-tailed Student’s *t* test was employed to assess the statistical significance between value pairs of different genotypes, resulting in *p* values. In this study, all error bars represent the mean ± standard error.

### 4.3. Quantitative RT-PCR and GUS Staining

The leaf explants after 4 HAC (hours after culture) on B5 medium were collected and snap-frozen in liquid nitrogen. Total RNA was isolated using the TRIZOL reagent (Invitrogen, Waltham, MA, USA), and cDNA was synthesized using a commercial kit (Thermo Scientific, Waltham, MA, USA). RT-qPCR was performed using the SYBR green supermix (Vazyme, Nanjing, China) on a Biorad CFX96 system (Biorad, Hercules, CA, USA). The RT-qPCR procedure and the primers for *CUC2*, *ABR1*, *ERF109*, *ASA1*, *ULT1*, *YUC1*, *YUC4*, *YUC6*, *YUC9*, and *eIF4A* (Appendix A) have been previously described [28,34,43,54]. All expression data represent the average of four biological replicates.

The detection of β-glucuronidase (GUS) activity in leaf explants was performed according to a published protocol with minor modifications [34,55]. In brief, leaf explants were harvested at specific time points after detachment (time 0 DAC and 1 DAC (day after culturing on B5 medium)) and immersed in the GUS assay solution (0.5 mM ferrocyanide, 0.5 mM ferricyanide, 50 mM NaHPO_4_, 50 mM Na_2_HPO_4_, and 1% Triton X-100) containing 1 mM X-Gluc (Solarbio, LIFE SCIENCE, Beijing, China). Then, the leaf explants in the GUS staining solution were vacuum-infiltrated for 30 min and subsequently incubated overnight at 37 °C. To remove the chlorophyll, the stained leaves were passed through an ethanol series and then photographed using a stereomicroscope (Olympus, Tokyo, Japan). Finally, the digital photographs were collated with Adobe Photoshop.

## Figures and Tables

**Figure 1 ijms-26-11336-f001:**
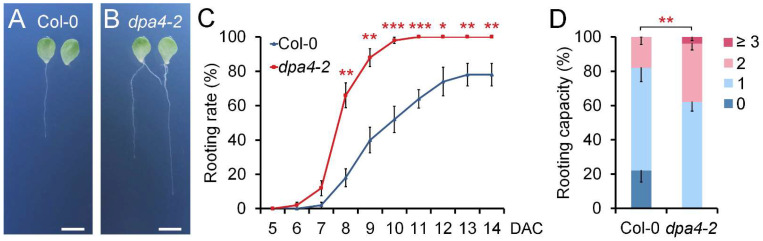
*DPA4* suppresses DNRR. (**A**,**B**) Root regeneration assay using wild-type (Col-0) (**A**) and *dpa4-2* (**B**) mutants on B5 medium at 14 DAC; scale bars indicate 500 µM. (**C**) Rooting rates in wild-type and *dpa4-2* mutant leaf explants on B5 medium under dark conditions. (**D**) Rooting capacity of leaf explants from wild-type and *dpa4-2* mutants, 14 DAC. 0, 1, 2, and ≥3 indicate the number of ARs per leaf explant. (**C**,**D**) Average values are shown (N ≥ 40 leaves from 5 individual plates), ±SEM. Asterisks indicate significant differences compared with Col-0 plants (Student’s *t*-test: * *p* < 0.05, ** *p* < 0.01, and *** *p* < 0.001).

**Figure 2 ijms-26-11336-f002:**
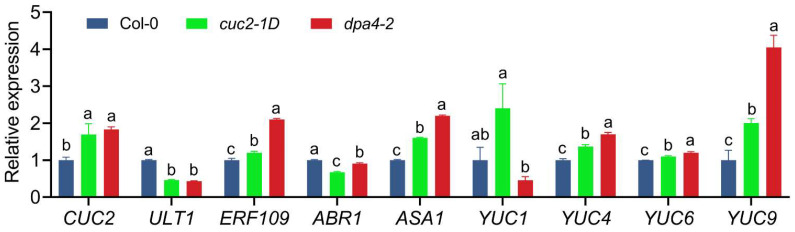
Gene expression levels in *cuc2-1D* and *dpa4-2* mutant leaf explants. RT-qPCR analysis of mRNA levels in wild-type, *cuc2-1D*, and *dpa4-2*, 4 h after culturing (4 HAC) on B5 medium. Average values are shown (N = 4), ±SEM. Statistical significance (*p* ≤ 0.05) was determined by one-way ANOVA and Duncan’s LSD, and a–c mark groups of significant differences.

**Figure 3 ijms-26-11336-f003:**
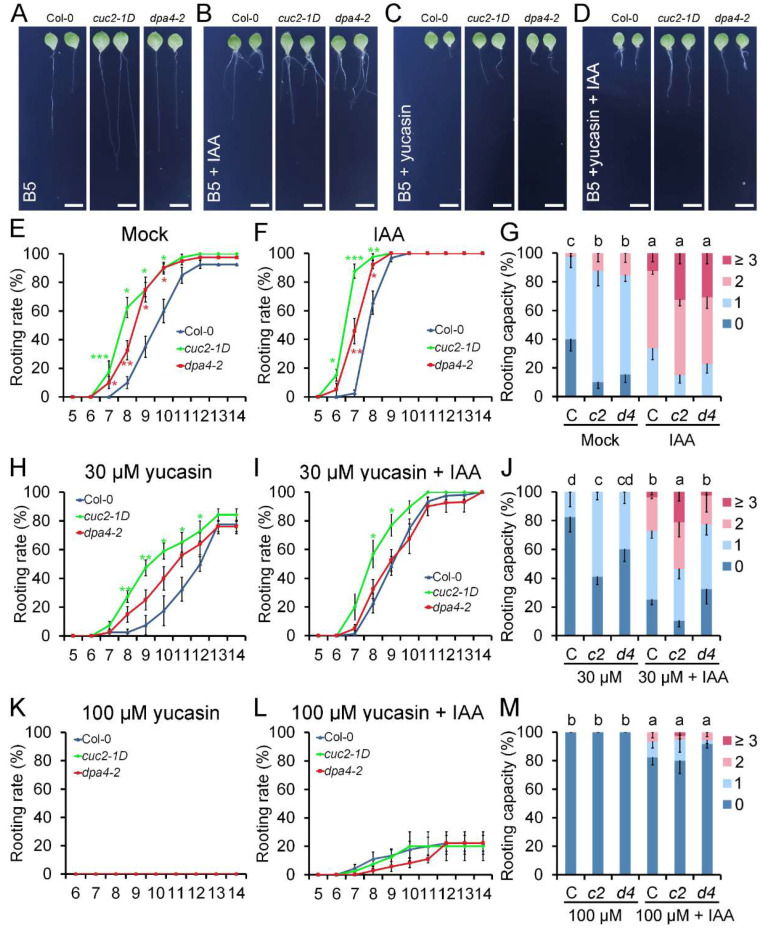
Auxin (IAA) and yucasin treatment during DNRR. (**A**–**D**) Leaf explants of Col-0, *cuc2-1D*, and *dpa4-2* cultured on B5 medium, 14 DAC. (**A**) mock; (**B**) 0.1 µM IAA; (**C**) 30 µM yucasin; (**D**) 30 µM yucasin + 0.1 µM IAA. Scale bars indicate 500 µm. (**E**,**F**) Rooting rate of Col-0, *cuc2-1D*, and *dpa4-2* on B5 medium; (**E**) mock and (**F**) 0.1 µM IAA. (**G**) Rooting capacity of leaf explants from Col-0 (C), *cuc2-1D* (*c2*), and *dpa4-2* (*d4*) on B5 medium, mock, and 0.1 µM IAA, 10 DAC. (**H**,**I**) Rooting rate of Col-0, *cuc2-1D*, and *dpa4-2* on (**H**) 30 µM yucasin and (**I**) 30 µM yucasin + 0.1 µM IAA. (**J**) Rooting capacity of leaf explants from Col-0 (C), *cuc2-1D* (*c2*), and *dpa4-2* (*d4*) on 30 µM yucasin and 30 µM yucasin + 0.1 µM IAA, 10 DAC. (**K**,**L**) Rooting rate of leaf explants from Col-0, *cuc2-1D*, and *dpa4-2* on (**K**) 100 µM yucasin and (**L**) 100 µM yucasin + 0.1 µM IAA. (**M**) Rooting capacity of leaf explants from Col-0 (C), *cuc2-1D* (*c2*), and *dpa4-2* (*d4*) on 100 µM yucasin and 100 µM yucasin + 0.1 µM IAA, 10 DAC. Average values are shown (N ≥ 40 leaves from 5 individual plates for each single experiment), ± SEM. Statistical significance was determined by Student’s *t*-test: * *p* < 0.05, ** *p* < 0.01, and *** *p* < 0.001, and a–d mark groups of significant differences (*p* ≤ 0.05). 0, 1, 2, and ≥3 indicate the number of ARs per leaf explant.

**Figure 4 ijms-26-11336-f004:**
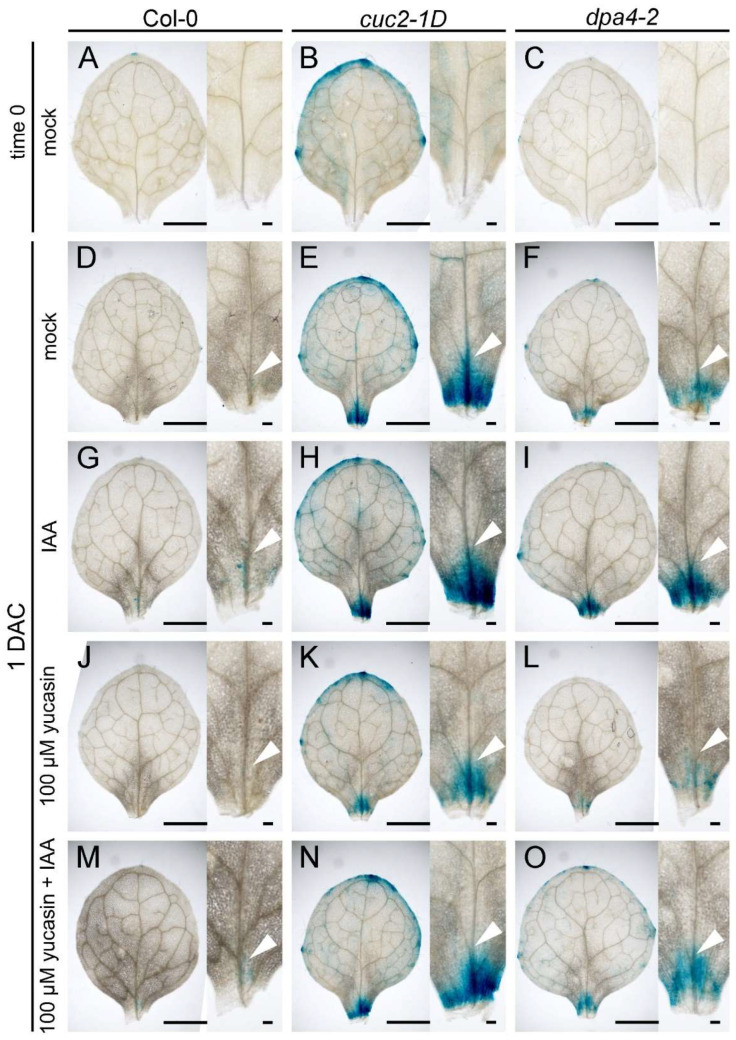
Auxin distribution in leaf explants affected by yucasin and IAA treatments. (**A**–**O**) GUS staining of leaf explants with *DR5::GUS* reporter in wild-type (Col-0; (**A**,**D**,**G**,**J**,**M**)), *cuc2-1D* (**B**,**E**,**H**,**K**,**N**), and *dpa4-2* (**C**,**F**,**I**,**L**,**O**) at 0 DAC (**A**–**C**) and 1 DAC (**D**–**O**). (**D**–**F**) Mock treatment at 1 DAC. (**G**–**I**) 0.1 μM IAA treatment at 1 DAC. (**J**–**L**) 100 μM yucasin treatment at 1 DAC. (**M**–**O**) 100 μM yucasin + 0.1 μM IAA treatment at 1 DAC. Arrowheads in (**D**–**O**) indicate the *DR5::GUS* signal in vasculature above the wounding side. Scale bars indicate 1000 µm and 100 µm.

**Figure 5 ijms-26-11336-f005:**
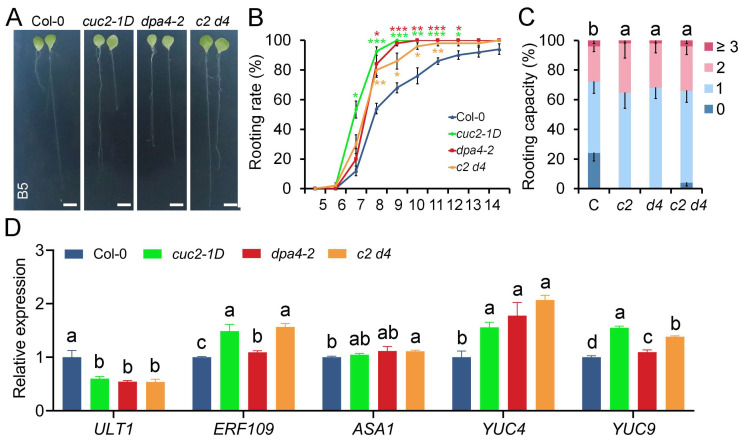
*DPA4* and *CUC2* regulate DNRR in a common genetic pathway. (**A**) Leaf explants cultured on B5 medium at 14 DAC from Col-0, *cuc2-1D*, *dpa4-2*, and *cuc2-1D dpa4-2*; scale bars indicate 500 µm. (**B**) Rooting rate of leaf explants from Col-0, *cuc2-1D*, *dpa4-2*, and *cuc2-1D dpa4-2* (*c2 d4*) on B5 medium. (**C**) Rooting capacity of leaf explants from Col-0 (C), *cuc2-1D* (*c2*), *dpa4-2* (*d4*), and *cuc2-1D dpa4-2* (*c2 d4*) on B5 medium at 10 DAC. (**D**) RT-qPCR analysis of gene expression levels in Col-0, *cuc2-1D*, *dpa4-2*, and *cuc2-1D dpa4-2* (*c2 d4*) leaf explants, 4 HAC on B5 medium, N = 4. (**B**,**C**) Average values are shown (N ≥ 40 leaves from 5 individual plates for each single experiment), ±SEM. Statistical significance was determined by Student’s *t*-test: * *p* < 0.05, ** *p* < 0.01, and *** *p* < 0.001 and a–d mark groups of significant differences (*p* ≤ 0.05). 0, 1, 2, and ≥3 represent AR numbers per leaf explant.

**Figure 6 ijms-26-11336-f006:**
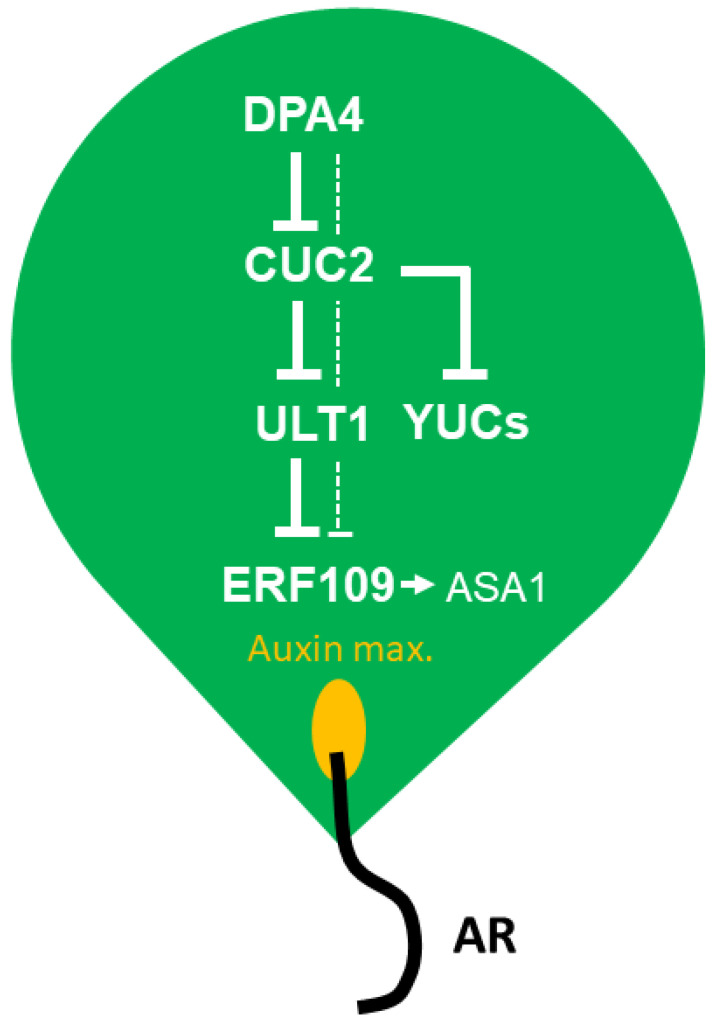
Conceptual model of the DPA4-CUC2-ULT1 module limiting auxin biosynthesis and adventitious root (AR) formation in Arabidopsis leaf explants. See the discussion for more details.

## Data Availability

The original contributions presented in this study are included in the article/Appendix A. Further inquiries can be directed to the corresponding author.

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
