# Peer review of "DPA4* Suppresses Adventitious Root Formation via Transcriptional Regulation of *CUC2* and *ULT1*, Decreasing Auxin Biosynthesis in Arabidopsis Leaf Explants"

_ijms, 2025, doi:10.3390/ijms262311336_

Round 1

Reviewer 1 Report

Comments and Suggestions for Authors

Comments

The introduction section is written well. I would like to recommend minor revision of the article. The author should strengthen the discussion section as per I suggested.

Line 97-110

These sentences are suitable for the results section not for introduction. And at this place better to write about the problems to be addressed purpose of the study and hypothesis etc.

Line 113 to 115 : kindly delete these sentences from the results section.

Line 153-154: The references are not needed in the results section. Better to only explain the findings of the current study.

Line 179 to 180: Either you explaining your findings or someone else. Kindly clearify.

The results are elaborated very well.

Line 277 delete “S” from the word factors

Line 280 replace effects to with effects on

Line 283 “controls likely auxin biosynthesis”  should be “likely controls auxin biosynthesis”

285         “a here newly identified promoter” should be “a newly identified promoter” or “a promoter newly identified here”

Line 288 “This findings” should be “These findings”

Line 322 “CUC2 and DPA4 regulates” should be “CUC2 and DPA4 regulate”

The discussion needs to be improved to strengthen the current findings

In the discussion author should use these statements to improve discussion with references.

Similar to how OsNLP6 in rice acts as a transcriptional regulator enhancing nitrogen use efficiency (Xin et al., 2025), DPA4 may function as a transcriptional repressor modulating auxin biosynthesis during DNRR. Genome-Wide Association Studies Identify OsNLP6 as a Key Regulator of Nitrogen Use Efficiency in Rice. Plant Biotechnology Journal. doi: https://doi.org/10.1111/pbi.70296

Line 327 “Conceptional model”should be “conceptual model”

Phosphorus and nitrogen cycling influence hormone metabolism and plant developmental responses (Sun et al., 2025), suggesting that nutrient–hormone crosstalk could further modulate DNRR efficiency.

Evidence for phosphorus cycling parity in nodulating and non-nodulating N2-fixing pioneer plant species in glacial primary succession. Functional Ecology, 39(4), 985-1000. doi: https://doi.org/10.1111/1365-2435.70023

Line 305 add comma before “suggesting”             

Line 332 “limit DNRR increasing the persistence”  better to write as “limit DNRR, thereby increasing the persistence…”

Line 317 add “is” before essential

JA-responsive transcription factors like PgMYB2 in Panax ginseng (Liu et al., 2019) demonstrate the importance of MeJA signaling in activating secondary pathways, supporting our observation that JA-induced ERF109 is involved in auxin-related DNRR regulation.

PgMYB2, a MeJA-Responsive Transcription Factor, Positively Regulates the Dammarenediol Synthase Gene Expression in Panax Ginseng. International Journal of Molecular Sciences, 20(9), 2219. https://doi.org/10.3390/ijms20092219

Line 309 “ULT1 may suppresses DNRR”should be “ULT1 may suppress DNRR”

Line 308 “ASA1-de-penent”  should be “ASA1-dependent”

Heat-responsive proteins such as TaHSP17.4 (Wang et al., 2023) illustrate how plants use molecular chaperones to improve regeneration and survival under stress, paralleling the protective role of transcriptional modules like DPA4–CUC2 in regulating post-injury root formation.

Heat shock protein TaHSP17.4, a TaHOP interactor in wheat, improves plant stress tolerance. International Journal of Biological Macromolecules, 246, 125694. doi: https://doi.org/10.1016/j.ijbiomac.2023.125694

Line 301 “depends on CUC2”  should be “depend on CUC2”

Line 341: delete of after 1%.

Line 347: delete the word “Plants”

Homeobox transcription factors are key regulators of organogenesis and regeneration, as shown by the WUSCHEL-related WOX family in Boehmeria nivea (Abubakar et al., 2023). Similarly, DPA4 and CUC2 coordinate root regeneration through transcriptional control of auxin biosynthetic genes.

Comprehensive Analysis of WUSCEL-Related Homeobox Gene Family in Ramie (Boehmeria nivea) Indicates Its Potential Role in Adventitious Root Development. Biology, 12(12), 1475. https://doi.org/10.3390/biology12121475

Materials and Methods

Line 339: replace surfaced with surface

Line 360: confirm the word “regent” or reagent ?

Line 361: change with “cDNA was synthesized using a commercial kit”

Line 374: replace with ““photographed using a stereomicroscope””

Author Response

First of all, we would like to thank all three reviewers for their detailed assessment of our manuscript. The constructive feedback of the reviewers has helped us to improve the quality of the manuscript and its potential impact in the field. We corrected all typos and followed all suggestions that improve our manuscript regarding to the editorial decision to accept our manuscript with minor revisions. Due to the time limitation for our revisions, some of the suggested experiments are far out of the scope of our manuscript, while some of these experiments would take over one year. Although some of the suggested experiments may improve our manuscript to a minor degree, none of them are essential for our presented results and/or our conclusions.

Please find below (and in the sections of the other reviewers) our point-to-point response. Please also check the revised manuscript for further changes.

Comments 1: The introduction section is written well. I would like to recommend minor revision of the article. The author should strengthen the discussion section as per I suggested.

Response 1: Thank you very much for your kind words about our writing. We are also very thankful for the suggested references and discussion points. However, none of the suggested studies focus on DNRR in Arabidopsis, only one study includes data to adventitious root development. Nevertheless, we see your point and revised our discussion wherever suitable to improve our manuscript (see revised manuscript).

Comments 2: Line 97-110 These sentences are suitable for the results section not for introduction. And at this place better to write about the problems to be addressed purpose of the study and hypothesis etc.

Response 2: We agree that the last paragraph of the introduction was lacking a proper addressing of purpose, hypothesis, and/or scientific question of the study. We therefore moved the first two sentences of the result part to the beginning of the last paragraph of the introduction. However, we did not delete the sentences in line 97-110, since they present a short summary of our results and conclusions, which gives a slightly different perspective in comparison to the abstract. To do so is common practice in scientific writing; and, in our opinion, is in line with the instruction of IJMS how to write the introduction.

Comments 3: Line 113 to 115: kindly delete these sentences from the results section.

Response 3: We deleted/moved the two sentence to the beginning of the last paragraph of the introduction.

Comments 4: Line 153-154: The references are not needed in the results section. Better to only explain the findings of the current study.

Response 4: Since our hypophysis based on our results and the literature the references are essential for the readers to understand that our reasoning is partially founded on published studies.

Comments 5: Line 179 to 180: Either you explaining your findings or someone else. Kindly clearify.

Response 5: We add 'Our' and the sentence read now 'Our treatment with 0.1 µM IAA accelerated ARs formation in all three genotypes without equalizing …'.

Comments 6: The results are elaborated very well.

Response 6: Thanks.

Comments 7: Line 277 delete “S” from the word factors

Response 7: Done!

Comments 8: Line 280 replace effects to with effects on

Response 8: Done!

Comments 9: Line 283 “controls likely auxin biosynthesis”  should be “likely controls auxin biosynthesis”

Response 9: Done!

Comments 10: 285         “a here newly identified promoter” should be “a newly identified promoter” or “a promoter newly identified here”

Response 10: Done!

Comments 11: Line 288 “This findings” should be “These findings”

Response 11: Done!

Comments 12: Line 322 “CUC2 and DPA4 regulates” should be “CUC2 and DPA4 regulate”

Response 12: Done!

Comments 13: The discussion needs to be improved to strengthen the current findings. In the discussion author should use these statements to improve discussion with references.

Response 13: Where appropriate, we revised the discussion (see revised manuscript).

Comments 14:  Similar to how OsNLP6 in rice acts as a transcriptional regulator enhancing nitrogen use efficiency (Xin et al., 2025), DPA4 may function as a transcriptional repressor modulating auxin biosynthesis during DNRR. Genome-Wide Association Studies Identify OsNLP6 as a Key Regulator of Nitrogen Use Efficiency in Rice. Plant Biotechnology Journal. doi: https://doi.org/10.1111/pbi.70296

Response 14: This reference is neither related to DNRR nor auxin synthesis.

Comments 15: Line 327 “Conceptional model”should be “conceptual model”

Response 15: Done!

Comments 16: Phosphorus and nitrogen cycling influence hormone metabolism and plant developmental responses (Sun et al., 2025), suggesting that nutrient–hormone crosstalk could further modulate DNRR efficiency.

Evidence for phosphorus cycling parity in nodulating and non-nodulating N2-fixing pioneer plant species in glacial primary succession. Functional Ecology, 39(4), 985-1000. doi: https://doi.org/10.1111/1365-243.705023

Response 16: This reference is neither related to DNRR nor auxin synthesis/signaling.

Comments 17: Line 305 add comma before “suggesting”    

Response 17: Done!

Comments 18: Line 332 “limit DNRR increasing the persistence”  better to write as “limit DNRR, thereby increasing the persistence…”

Response 18: Done!

Comments 19: Line 317 add “is” before essential

Response 19: Done!

Comments 20:  JA-responsive transcription factors like PgMYB2 in Panax ginseng (Liu et al., 2019) demonstrate the importance of MeJA signaling in activating secondary pathways, supporting our observation that JA-induced ERF109 is involved in auxin-related DNRR regulation.

PgMYB2, a MeJA-Responsive Transcription Factor, Positively Regulates the Dammarenediol Synthase Gene Expression in Panax Ginseng. International Journal of Molecular Sciences, 20(9), 2219. https://doi.org/10.3390/ijms20092219

Response 20: Although JA is a key regulator of DNRR by directly activating ERF109/111, this reference is neither related to DNRR nor auxin synthesis/signaling.

Comments 21: Line 309 “ULT1 may suppresses DNRR”should be “ULT1 may suppress DNRR”

Response 21: Done!

Comments 22: Line 308 “ASA1-de-penent”  should be “ASA1-dependent”

Response 19: The correct hyphenation of “ASA1-depenent” is an automatic function of the IJMS word template file.

Comments 23:  Heat-responsive proteins such as TaHSP17.4 (Wang et al., 2023) illustrate how plants use molecular chaperones to improve regeneration and survival under stress, paralleling the protective role of transcriptional modules like DPA4–CUC2 in regulating post-injury root formation.

Heat shock protein TaHSP17.4, a TaHOP interactor in wheat, improves plant stress tolerance. International Journal of Biological Macromolecules, 246, 125694. doi: https://doi.org/10.1016/j.ijbiomac.2023.125694

Response 16: This reference is neither related to DNRR nor auxin synthesis. There is also no evidence that DPA4 or CUC2 interact with any chaperone or that this interaction is related to their function as transcription factors.

Comments 24: Line 301 “depends on CUC2”  should be “depend on CUC2”

Response 24: Done!

Comments 25: Line 341: delete of after 1%.

Response 25: Done!

Comments 26: Line 347: delete the word “Plants”

Response 26: Done!

Comments 27:  Homeobox transcription factors are key regulators of organogenesis and regeneration, as shown by the WUSCHEL-related WOX family in Boehmeria nivea (Abubakar et al., 2023). Similarly, DPA4 and CUC2 coordinate root regeneration through transcriptional control of auxin biosynthetic genes.

Comprehensive Analysis of WUSCEL-Related Homeobox Gene Family in Ramie (Boehmeria nivea) Indicates Its Potential Role in Adventitious Root Development. Biology, 12(12), 1475. https://doi.org/10.3390/biology12121475

Response 17: Thank you to point out the importance of the WOX gene family for DNRR. As described in our introduction, WOX11/12 and WOX5 are downstream targets of the 'auxin maxima' during DNRR, and therefore downstream of DPA4 and CUC2. We add WOX11 expression in Supplementary Figure S3 and in the discussion the sentence in line 322 " Furthermore, the auxin-inducible DNRR-factor WOX11 was significantly upregulated in dpa4-2, cuc2-1D, and miR164a-4 (Supplementary Figure S3) indicating an link between increased CUC2 expression and auxin biosynthesis, but more importantly, an immediate increase of a key regulator of cell fate reprograming during DNRR, WOX11 [23], [24]."

Comments 28: Materials and Methods Line 339: replace surfaced with surface

Response 28: Done!

Comments 29: Line 360: confirm the word “regent” or reagent ?

Response 29: Done!

Comments 31: Line 361: change with “cDNA was synthesized using a commercial kit”

Response 31: Done!

Comments 32: Line 374: replace with ““photographed using a stereomicroscope””

Response 32: Done!

Reviewer 2 Report

Comments and Suggestions for Authors

In the manuscript “A loss of DPA4 mutant reveals that the boundary gene CUC2 promotes adventitious root formation in Arabidopsis leaf explants via repression of ULT1 and increased Auxin biosynthesis”, the authors attempted to reveal a novel mechanism of the DPA4–CUC2–ULT1 module in regulating adventitious root regeneration. There are some problems with the author's writing and description in the manuscript:

  1. The title is lengthy, suggest to revise it.
  2. No auxin contentto confirm reduced auxin levels in dpa4-2.Increasing the determination of auxin content can enhance the reliability of the results.
  3. Yucasin may have off-target effects, genetic validation is needed.

Author Response

First of all, we would like to thank all three reviewers for their detailed assessment of our manuscript. The constructive feedback of the reviewers has helped us to improve the quality of the manuscript and its potential impact in the field. We corrected all typos and followed all suggestions that improve our manuscript regarding to the editorial decision to accept our manuscript with minor revisions. Due to the time limitation for our revisions, some of the suggested experiments are far out of the scope of our manuscript, while some of these experiments would take over one year. Although some of the suggested experiments may improve our manuscript to a minor degree, none of them are essential for our presented results and/or our conclusions.

Please find below (and in the sections of the other reviewers) our point-to-point response. Please also check the revised manuscript for further changes.

Comments 1: In the manuscript “A loss of DPA4 mutant reveals that the boundary gene CUC2 promotes adventitious root formation in Arabidopsis leaf explants via repression of ULT1 and increased Auxin biosynthesis”, the authors attempted to reveal a novel mechanism of the DPA4–CUC2–ULT1 module in regulating adventitious root regeneration. There are some problems with the author's writing and description in the manuscript: 1. The title is lengthy, suggest to revise it.

Response 1: We shortened the title that reads now "DPA4 suppresses adventitious root formation via transcriptional regulation of CUC2 and ULT1 decreasing auxin biosynthesis in Arabidopsis leaf explants"

Comments 2: No auxin content to confirm reduced auxin levels in dpa4-2. Increasing the determination of auxin content can enhance the reliability of the results.

Response 2: Although we agree that measurements of the auxin content may enhance the reliability of our results, we think that these auxin data are not required for the following reasons: (i) Our DR5::GUS assays clearly showed increased auxin responses in dpa4-2 and cuc2-1D mutants, while the DR5::GUS assay is a commonly used method to measure changes in auxin content and auxin location. (ii) Our expression analyses evidently pointed to increased auxin biosynthesis caused by increased expression levels of auxin biosynthesis genes. (iii) Taken together, increased auxin response (DR5::GUS) and increased levels of several auxin biosynthesis genes (RT-qPCR) obviously pointed to increased auxin biosynthesis as main factor of the observed increase of adventitious root formation in dpa4-2 and cuc2-1D mutant leaf explants. (iv) Furthermore, the measurement of the auxin content would be very difficult in the tiny Arabidopsis leaf explants, and an expensive approach as well, while the DR5::GUS assay is very reliable and used in thousands of plant studies including high impact papers.

Comments 3: Yucasin may have off-target effects, genetic validation is needed.

Response 3: We fully agree that it is very likely that Yucasin has off-target effects. Nevertheless, DR5::GUS staining with and without Yucasin (& IAA) treatment strongly correlates with the rooting rate and rooting capacity. Therefore, changes in auxin biosynthesis seem the main factor that determines rooting rate and rooting capacity. Furthermore, genetic validation would require at least 1-2 years, since several auxin biosynthesis genes are misexpressed in dpa4-2, cuc2-1D, and miR164a-4 mutants. Here, the required effort is incommensurate with the expectedly low increase of knowledge.

Reviewer 3 Report

Comments and Suggestions for Authors

The authors tried to characterize mutants and stated that DPA4 enhances root initiation through CUC2-mediated repression of ULT1 and stimulation of auxin biosynthesis in Arabidopsis leaf explants. The authors need to minimize plagiarism similarity throughout the text. The following points should be considered:

Lines 28-30: To conclude this sentence, the author needs to analyze chromatin immunoprecipitation (ChIP-seq) to test whether binding occurs at CUC2 and ULT1 promoter regions or not. And need transcriptional activation/repression assays to determine whether DPA4 reduces LUC from proCUC2 and increases (or causes no repression of) proULT1 LUC activity.

Lines 121-122: In Figure 1A, why does one Col-0 sample plant have no roots? In biological sample analysis (Fig A and B), the author needs to show at least 3 plant phenotypes.  

123-124: Maybe DPA4 plays a role in suppressing DNRR in Arabidopsis. Without further confirmation, nobody can surely say that this root phenotype is produced due to the lack of DPA4 gene.

The author needs to write a clear conclusion for the present findings.

The author needs to check the reference lists of those that are cited in the text. All the titles and other contents should be unique to the journal format.

Author Response

First of all, we would like to thank all three reviewers for their detailed assessment of our manuscript. The constructive feedback of the reviewers has helped us to improve the quality of the manuscript and its potential impact in the field. We corrected all typos and followed all suggestions that improve our manuscript regarding to the editorial decision to accept our manuscript with minor revisions. Due to the time limitation for our revisions, some of the suggested experiments are far out of the scope of our manuscript, while some of these experiments would take over one year. Although some of the suggested experiments may improve our manuscript to a minor degree, none of them are essential for our presented results and/or our conclusions.

Please find below (and in the sections of the other reviewers) our point-to-point response. Please also check the revised manuscript for further changes.

Comments 1: The authors tried to characterize mutants and stated that DPA4 enhances root initiation through CUC2-mediated repression of ULT1 and stimulation of auxin biosynthesis in Arabidopsis leaf explants. The authors need to minimize plagiarism similarity throughout the text. The following points should be considered:

Lines 28-30: To conclude this sentence, the author needs to analyze chromatin immunoprecipitation (ChIP-seq) to test whether binding occurs at CUC2 and ULT1 promoter regions or not. And need transcriptional activation/repression assays to determine whether DPA4 reduces LUC from proCUC2 and increases (or causes no repression of) proULT1 LUC activity.

Response 1: We fully agree that ChIP-seq could indicate whether DPA4 binding occurs at the promoter regions of CUC2 and ULT1 or not. However, such experiments are simply beyond the scope of our manuscript. Since there is no evidence that DPA4 regulates the expression of CUC2 and ULT1 directly, no transcriptional activation/repression assays are required.

Comments 2: Lines 121-122: In Figure 1A, why does one Col-0 sample plant have no roots? In biological sample analysis (Fig A and B), the author needs to show at least 3 plant phenotypes.  

Response 2: Pictures of biological samples should illustrate the static data and show examples of some but not all phenotype varieties. Nevertheless, we think that Figure 1A and 1B are reflecting very well the rooting rate and rooting capacity data of Figure 1C and 1D. The acceleration of rooting in dpa4-2, which is demonstrated in the earlier increase of the rooting rate (1C), is well illustrated by the longer roots of dpa4-2 (which emerged earlier; 1B) compared to the shorter roots of Col (1A). Both Col leaf explants have roots (1A); but even not, about 20% of all Col leaf explants have no roots at 14 DAC (1C-D). Furthermore, one of our examples of dpa4-2 leaf explants have two roots that reflects the increased rooting capacity of dpa4-2 mutants compared to Col (1D).

Comments 3: 123-124: Maybe DPA4 plays a role in suppressing DNRR in Arabidopsis. Without further confirmation, nobody can surely say that this root phenotype is produced due to the lack of DPA4 gene.

Response 3: The loss-of-function mutant dpa4-2 is an established strong T-DNA allele (Engelhorn et al., 2012). There is no evidence for any background mutants or additional T-DNA insertions in this plant line. Therefore, it is most likely that the rooting phenotype is due to the lack of the DPA4 gene. By using an established mutant allele, it would be very unusual to perform further controls.

Comments 4: The author needs to write a clear conclusion for the present findings.

Response 4: A conclusion section is not mandatory in IJMS and should only be added to the manuscript if the discussion is unusually long or complex. In our opinion, our discussion is neither unusually long nor complex, but clearly describes how we drew our conclusions from our results and the literature; we also discuss the limitations of our results and conclusions.

The author needs to check the reference lists of those that are cited in the text. All the titles and other contents should be unique to the journal format.

Response 1: We checked the reference lists and citations in the text and made two corrections:

We add the doi number to reference [3] and replaced (Ye et al, 2020) in line 91 by [11].